# Accurate and fast detection of copy number variations from short-read whole-genome sequencing with deep convolutional neural network

## Abstract

A copy number variant (CNV) is a type of genetic mutation where a stretch of DNA is lost or duplicated once or multiple times. CNVs play important roles in the development of diseases and complex traits. CNV detection with short-read DNA sequencing technology is challenging because CNVs significantly vary in size and are similar to DNA sequencing artifacts. Many methods have been developed but still yield unsatisfactory results with high computational costs. Here, we propose CNV-Net, a novel approach for CNV detection using a six-layer convolutional neural network. We encode DNA sequencing information into RGB images and train the convolutional neural network with these images. The fitted convolutional neural network can then be used to predict CNVs from DNA sequencing data. We benchmark CNV-Net with two high-quality whole-genome sequencing datasets available from the Genome in a Bottle Consortium, considered as gold standard benchmarking datasets for CNV detection. We demonstrate that CNV-Net is more accurate and efficient in CNV detection than current tools.

## 1 Introduction

A copy number variant (CNV) is a genetic mutation where a stretch of DNA is completely lost or repeated more than once compared to a reference genome. CNV sizes range from 50 bases to 3 million bases or more with two major types: duplication if the DNA sequence is repeated and deletion if a DNA sequence is missing. CNVs are spread along a genome, which account for 4.8% to 9.5% of human genome (Zarrei et al., 2015). They are known to influence many complex diseases including autism (Sebat et al., 2007), bipolar disorder (Green et al., 2016), schizophrenia (Stefansson et al., 2008), and Alzheimer' disease (Cuccaro et al., 2017) as well as gene expression (Chiang et al., 2017). As CNVs may overlap with a large portion of a genome, such as one gene or even multiple genes, their effect on disease may be substantially larger than that of single nucleotide variants, and hence CNVs often play important roles in the genetic mechanisms of diseases and complex traits.

With the advent of next-generation DNA sequencing technologies over the past decade, the resolution and scale of CNV detection has been greatly improved as large-scale sequencing studies become feasible. However, CNVs are still challenging to detect from short-read next-generation DNA sequencing techniques due to the significantly varying sizes of CNVs and their similarity with common DNA sequencing artifacts. Many computational methods have been developed for the discovery of CNVs from short-read DNA sequencing data, but their performance is often unsatisfactory due to low accuracy and high computational cost (Kosugi et al., 2019). The main reason for this is that previous methods (Rausch et al., 2012; Abyzov et al., 2011) mainly rely on the statistical analysis of the signals from read alignments (a process that maps reads from DNA sequencing data to the reference genome), which often fail to employ all useful features of DNA sequencing data and require significant computational resources. Therefore, there is a need for a novel, sophisticated computational tool to improve CNV detection with higher accuracy and efficiency.

Here, we present CNV-Net, a new approach to identify CNVs from DNA sequencing data using a six-layer deep convolutional neural network (CNN). CNV-Net first encodes the reads and their

related information from DNA sequencing data such as the type of base, base coverage (the number of reads covering a position), and base quality of each position into the RGB channels of images. Then it uses a deep CNN to predict these images as deletions, duplications, or false positives (not a CNV) breakpoints. Note that a CNV is defined by two breakpoints, that is, its start and end positions. Although it has been shown that CNNs such as DeepVariant (Poplin et al., 2018) and Clairvoyante (Luo et al., 2019) can accurately detect single nucleotide variants from similarly encoded images or tensors, few CNNs have been designed for the identification of CNVs. The key advantage of CNV-Net over other methods is the application of deep learning, which allows learning features with higher complexity from DNA sequencing data. We train the network with the Genome in a Bottle Consortium NA12878 dataset (Pendleton et al., 2015), and demonstrate CNV-Net performance on a high-quality benchmarking dataset acquired from a well-studied cell line, HG002 (Zook et al., 2020). We also compare its performance to that of previous methods.

## 2 METHODS

CNV-Net first encodes sequencing information into pileup images and classifies each image as a deletion, duplication, or false positive (not a CNV). In this section, we describe how we generate pileup images, that is, the input to CNV-Net, as well as the design of the CNN.

### 2.1 PILEUP IMAGE GENERATION

We convert the information of mapped reads from DNA sequencing data (provided in BAM file format) and reference genome information to 201×55 pixel image representation of 201-base wide regions. Training data pileup images contain a CNV breakpoint (start or end position) centered in the image, capturing the 100 positions left and right each CNV breakpoint. Using the RGB channels of an image, this snapshot is able to capture features of the reads at specific locations in the reference genome.

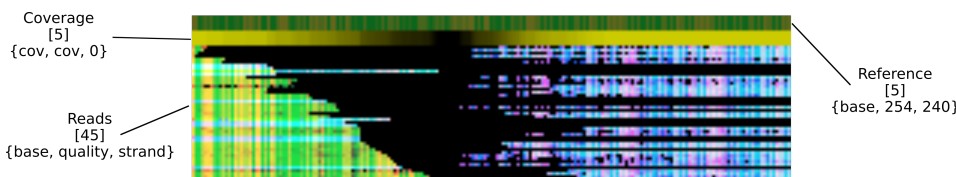

Figure 1: 201×55 pixel pileup image. Feature rows are labeled by brackets and RGB channels are labeled by curly braces. Reference rows are encoded in R channel by base A: 250, G: 180, T: 100, C: 30, N: 0. Coverage rows are encoded in RG channels by the number of mapped reads covering each position. Reads are encoded in RGB channels by base types, base quality, and strand directionality positive: 70, negative: 240.

As shown in Figure 1, the first five rows of the image capture the reference genome across a region of 201 base pairs by encoding the R channel of each pixel with an individual base in that position: A's as 250, G's as 180, T's as 100, C's as 30, and N's as 0. Under the five rows representing the reference genome are another five rows that capture the read depth (coverage) at that position. The R and G channels are encoded with an adjusted read depth value between 0 and 255. The following 45 rows capture the mapped reads from DNA sequencing. The R channel encodes for the nucleotide, which is the same as the reference genome, and the G channel encodes for the base quality found in the BAM file, which is adjusted to be between 0 and 255. Lastly, the B channel encodes for strand directionality: 70 if positive and 240 if negative.

Each 201×55 pixel pileup image, thus, captures coverage and reads specific to the 201-base region of the reference genome such that the sequencing data may be re-created from a series of consecutive pileup images. If there are more than 45 reads mapping to a specific region, those reads are discarded; we instead compensate by capturing this information through the rows that encodes coverage. If no reads map to the region, the image is left blank below the first five rows for the reference genome.

## 2.2 DESIGN OF CONVOLUTIONAL NEURAL NETWORK

Figure 2: CNV-Net architecture. CNV-Net employs a six-layer convolutional neural network for detecting copy number variations from DNA sequencing data. Descriptions under each layer indicate: type of layer, dimension of layer in parenthesis (input layer: height × width × channels, convolutional layer: height × width × filters, and fully connected layer: nodes), kernel size in brackets. The model predicts the probability of deletion ("DEL"), duplication ("DUP"), and false positive ("FP") breakpoint presence in each input pileup image.

As presented in Figure 2, CNV-Net consists of four convolutional layers and two fully connected layers with a total of 5,110,979 parameters. Each of the four convolutional layers uses $3 \times 3$ kernels and the RELU activation function. Each convolutional layer is followed by batch normalization, average pooling, and 0.4 dropout. Arrays are flattened in between the four convolutional layers and two fully connected layers followed by softmax output to predictions for CNV breakpoint presence in the 201-base window: duplication, deletion, or false positive (not a CNV).

In training CNV-net, we use a standard Adam optimizer (Kingma & Ba, 2014) with a sparse categorical cross-entropy loss function. We use a cyclical learning rate initialized at 1e-5 and maximal 1e-2. To reduce overfitting, we train the model until the accuracy cannot be improved in 15 epochs. The 201-base regions are centered around the CNV breakpoints, that is, the starts and ends of CNVs in the training set capturing the 100 positions directly before and after each CNV breakpoint. False positive images include random 201-base regions outside of CNVs. From the NA12878 dataset, a total of 52,243 pileup images are labeled as duplications, deletions, and false positives from the training set randomly split into 70%, 20%, 10% categories for training set, validation set, and test set for the network, respectively.

## 3 RESULTS

In this study, we use two standard structural variation benchmarking datasets: NA12878 and HG002. Both these benchmarking datasets are mapped to the GRCh37 reference genome. NA12878 is a well-studied genome provided by the Genome in a Bottle Consortium. It is whole-genome sequenced with $50\times$ coverage and each read in the sequencing data is 101 bases long. From the NA12878 image dataset (Pendleton et al., 2015), we generate images for 22,936 duplications, 18,614 deletions, and 10,693 false positives (not CNVs) breakpoints. The HG002 dataset (Zook et al., 2020) is another robust benchmarking dataset derived by the Genome in a Bottle Consortium, which is sequenced with $60\times$ coverage. Reads in the HG002 dataset are all 148 bases long. As we only consider CNVs passing the quality control filter, this dataset contains 174 duplications, 10,668 deletions, and 5,205 false positives breakpoints post-filtering.

We train the convolutional neural network on the NA12878 dataset and test the fitted model on the HG002 dataset. We also compare CNV-Net with popular existing tools for CNV detection, including BreakDancer (Chen et al., 2009), CNVnator (Abyzov et al., 2011), Delly (Rausch et al.,

2012), Lumpy (Layer et al., 2014), and Manta (Chen et al., 2016). It is important to note that CNV-Net makes predictions using the pileup images of CNVs where their start and end positions are given, and hence we assume that candidate CNV positions are given as input to CNV-Net. However, other methods find the candidate CNV positions using the information on how reads are mapped to the reference genome and detect CNVs on those candidate CNV positions.

The NA12878 dataset is preprocessed and converted into RGB images as described in the Methods section. We split the CNVs in NA12878 into 70%, 20%, 10% for training set, validation set, and test set, respectively. We train the CNV-Net with the training set and report its performance metrics on the test set. As shown in Table 1, CNV-Net identifies duplications with F1-score of 0.7833, deletions with F1-score of 0.7473, and false positives with F1-score of 0.7124. This result shows that CNV-Net achieves high accuracy in the test set of the NA12878 dataset.

Table 1: Performance of CNV-Net on the test set of the NA12878 dataset

| CNV types | Precision | Recall | F1-score |
|---|---|---|---|
| Duplications | 0.8376 | 0.7356 | 0.7833 |
| Deletions | 0.7885 | 0.7102 | 0.7473 |
| False positives | 0.6046 | 0.8672 | 0.7124 |

Next, we evaluate our fitted CNV-Net on the HG002 dataset and compare it with five existing tools for CNV detection. This training scheme represents the real data analysis as we will train CNV-Net with CNVs from known genomes and testing it on different genomes. As demonstrated in Table 2, CNV-Net achieves the highest F1-score in CNV detection in the HG002 dataset among all six methods. It also has the highest recall and the second highest precision. This result shows that CNV-Net is more accurate in CNV detection than current methods.

Table 2: Performance of CNV-Net and other methods on the HG002 dataset

| Tools | Precision | Recall | F1-score |
|---|---|---|---|
| BreakDancer | 0.05437 | 0.6503 | 0.05923 |
| CNVnator | 0.3523 | 0.03143 | 0.05771 |
| Delly | 0.5841 | 0.06846 | 0.1226 |
| Lumpy | 0.7228 | 0.06815 | 0.1245 |
| Manta | 0.9282 | 0.3916 | 0.5508 |
| **CNV-Net** | **0.7960** | **0.6596** | **0.6912** |

We then compare the time cost of applying different methods to the HG002 dataset. CNV-Net is significantly faster than other methods (Table 3). That is because a fitted convolutional neural network is efficient in making predictions, however, it may be time-consuming during model training. Run time is effectively front-loaded into model training instead of a concurrent statistical analysis. Another reason for this can be that CNV-Net makes predictions on candidate CNVs so it does not need to spend time searching for candidate CNVs in DNA sequencing data as if other methods do.

Table 3: Runtime of CNV-Net and other methods on the HG002 dataset

| Tools | Runtime |
|---|---|
| BreakDancer | 92 minutes |
| CNVnator | 51 minutes |
| Delly | 565 minutes |
| Lumpy | 46 minutes |
| Manta | 36 minutes |
| **CNV-Net** | **16.62 seconds** |

## 4   CONCLUSION

CNV-Net encodes DNA sequencing information into RGB images and applies a deep convolutional neural network to learn complex features from DNA sequencing data to discover CNVs. We demonstrate that CNV-Net can detect CNVs breakpoints from DNA sequencing data with significantly higher accuracy and efficiency than existing tools.

To the best of our knowledge, CNV-Net is the first tool to use a CNN to detect CNVs from DNA sequencing data and achieve high performance on the standard benchmarks. It changes the problem of CNV detection from a process of statistical modeling requiring professional expertise into a process of optimizing a general deep learning model. Besides, although we benchmark CNV-Net using only human samples in this study, it can also be applied to DNA sequencing data of other species.

One limitation of CNV-Net is its need to know candidate CNV positions, potential start and end positions of CNVs, before applying the CNN. This is because CNV-Net makes predictions on pileup images generated at the start and end positions of candidate CNVs. In our analysis, we used candidate CNV positions in human genomes, which were previously identified by the Genome in a Bottle Consortium using a suite of other algorithms. In genome sequencing datasets where CNVs have not been analyzed, these candidate CNV positions may not be available. One solution to this problem may be using candidate CNV positions detected by other CNV detection algorithms, especially algorithms with high recall, as CNV-Net can remove false positive CNVs from those CNV positions. Additionally, we may also develop a sensitive candidate CNV detection algorithm, which may have low precision but high recall. Our future research direction involves identifying candidate CNV positions from genome sequencing datasets and incorporating them with CNV-Net; nevertheless, CNV-Net demonstrates potential as a valuable and effective tool for CNV detection.

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

Justin M Zook, Nancy F Hansen, Nathan D Olson, Lesley Chapman, James C Mullikin, Chunlin Xiao, Stephen Sherry, Sergey Koren, Adam M Phillippy, Paul C Boutros, Sayed Mohammad E Sahraeian, Vincent Huang, Alexandre Rouette, Noah Alexander, Christopher E Mason, Iman Hajirasouliha, Camir Ricketts, Joyce Lee, Rick Tearle, Ian T Fiddes, Alvaro Martinez Barrio, Jeremiah Wala, Andrew Carroll, Noushin Ghaffari, Oscar L Rodriguez, Ali Bashir, Shaun Jackman, John J Farrell, Aaron M Wenger, Can Alkan, Arda Soylev, Michael C Schatz, Shilpa Garg, George Church, Tobias Marschall, Ken Chen, Xian Fan, Adam C English, Jeffrey A Rosenfeld, Weichen Zhou, Ryan E Mills, Jay M Sage, Jennifer R Davis, Michael D Kaiser, John S Oliver, Anthony P Catalano, Mark J P Chaisson, Noah Spies, Fritz J Sedlazeck, and Marc Salit. A robust benchmark for detection of germline large deletions and insertions. *Nat. Biotechnol.*, June 2020.

