# OpenReview forum: "Accurate and fast detection of copy number variations from short-read whole-genome sequencing with deep convolutional neural network"
_ICLR.cc/2021/Conference — Reject_

### Official Review · AnonReviewer1 · 2020-10-27
**Official Blind Review #1**

**Rating:** 3
**Confidence:** 4

**Review:**

##########################################################################

Summary:
The authors proposed CNV-Net, a deep learning-based approach for copy number variation identification. They encoded mapped DNA sequences into a pileup image that captures reference sequence, sequencing coverage, and mapped reads. Then, they used CNNs to classify it into deletions, duplications, or non-breakpoints. They benchmarked CNV-Net with two whole-genome sequencing datasets and claimed to obtain more accurate and efficient results than current tools.


##########################################################################

Major comments:
While the paper has its own merits, unfortunately, it has several issues that need to be addressed.
- Although the authors claimed that CNV-Net is the first tool to use a CNN to detect CNVs, this is not true. Several previous works used CNNs for CNV detection. Furthermore, I think other machine learning and deep learning-based works should also be acknowledged. I’d recommend authors properly cite and compare previous works with the proposed method. Some of the previous works include the followings:
(1) Zhang, Yun Xiang, et al. "DL-CNV: A deep learning method for identifying copy number variations based on next generation target sequencing." Mathematical biosciences and engineering: MBE 17.1 (2019): 202-215.
(2) Cai, Lei, Yufeng Wu, and Jingyang Gao. "DeepSV: accurate calling of genomic deletions from high-throughput sequencing data using deep convolutional neural network." BMC bioinformatics 20.1 (2019): 665.
(3) Hill, Tom, and Robert L. Unckless. "A deep learning approach for detecting copy number variation in next-generation sequencing data." G3: Genes, Genomes, Genetics 9.11 (2019): 3575-3582.
(4) Pounraja, Vijay Kumar, et al. "A machine-learning approach for accurate detection of copy number variants from exome sequencing." Genome research 29.7 (2019): 1134-1143.
- The main contribution of the paper would be using a pileup image of mapped reads and a CNN to detect CNVs. However, in my view, the novelty of the paper is quite limited. As stated in the introduction, the pileup image encoding and CNNs have already been used in a couple of previous works for SNV detection. I could find any significant methodological differences in CNV and SNV detections; it seems like rather a straightforward application of previous methods on another similar problem. Otherwise, please clarify the differences between the two problems and what authors have done to overcome the new obstacles.
- The core issue I have with this paper is that I do not think the experiment settings are realistic. As stated by the authors, CNV-Net must know the candidate CNV positions. I think this is a serious issue that must be handled rather than leaving it as a limitation of the work. Currently, the CNV-Net is evaluated with mapped and pre-preprocessed reads with CNV centered breakpoints. Compared to the experiments conducted in the previous works, the experiments of the proposed work seem limited, unrealistic, and biased in favor of the proposed method.
- In my view, the authors left out too much information. For examples, it is quite difficult to understand how they used other tools for the experiments; Do they only use the CNV breakpoints that passed the quality control filter as CNV-Net? Or do they use all the mapped reads? What tool-specific arguments did the authors use for each tool? Currently, it is extremely difficult to reproduce the results presented in the paper.

##########################################################################

Minor comments:
- How did the authors choose the specific numbers to encode individual base into R channel (e.g. A with 250, G with 180)
- In the results section, the authors stated that they only used CNVs passing the quality control filter. Please provide more details for the filter explaining the filtering criteria and how they chose them.
- In Table 2, how did the authors obtain the metrics for the multi-class problem? Please state whether they are macro or micro averages of scores.

##########################################################################

---

### Official Review · AnonReviewer3 · 2020-10-27
**Odd strategy, insufficiently described benchmark**

**Rating:** 2
**Confidence:** 4

**Review:**

__Summary__
They authors describe a method to detect structural variation from aligned sequencing reads in a genome browser view. Their model encodes this genome browser view into an RGB image and applies a deep convolutional neural network to classify variant type (or no variant). They make use of curated variant annotations to train and test their model.

__Major comments__
* The RGB encoding is entirely arbitrary, unnecessary, and confusing. The authors should consider the actual range of the various input data and encode with a simple and interpretable strategy. For example, nucleotides are typically one hot encoded.

* The improved accuracy on this task is abstract when only summarized in tables. Depicting an example of a structural variant whose prediction is improved by the authors’ method would be very valuable. Ideally, both a false positive turned true negative and a false negative turned true positive could be shown.

* The requirement that this method be provided candidate structural variation start and end points means that it's actually a module that would need to be plugged into a pipeline that also specified how those candidates are obtained. I would encourage the authors to develop that strategy before publishing their method.

* The authors have not clearly described how the predictions of the other methods were used to annotate these curated variants. Doing so involves critical parameters, such as the allowed distance between the method prediction and true specified variant break points. Setting these parameters to strict values would be very unfair given that the authors method is not required to produce such breakpoints de novo. The Zook et al. paper describes this process in detail and discusses several software packages to do it.

__Minor comments__
* Does the authors’ method make use of paired end read information?
* How does Table 2 combine the accuracies for duplications and indels?

---

### Official Review · AnonReviewer4 · 2020-10-28
**A very short, trival paper, recommend rejection**

**Rating:** 2
**Confidence:** 5

**Review:**

The authors in the paper describe a deep learning approach to detect copy number variants (CNVs) from DNA sequencing data, CNV-Net.  It described the approach by transforming the pileups into images and pass them through a CNN. This strategy has been proposed four or five years back to do SNPs and indels calling (DeepVariant).  It is challenging for SVs and CNVs as they can be arbitrarily large. However, there are also several existing models for this task (e.g. DeepSV, RDBKE). In this work, the authors only consider "candidate CNV regions" which are 201-bp small genomic regions that centered at the breakpoints.

The paper is only 4.5 pages. First, the authors did not explain or investigate many decisions in their model design. Then the experiments are flawed. I have many questions and concerns.
1) First of all, this should not be called a CNV detector because it is in fact a breakpoint detector
2) How did the authors do negative sampling? Were the negative samples randomly drawn from the genome? This creates bias as the sequence features might shift dramatically.
3) Why are the negative samples not balanced with the positive samples?
4) Are there differences in performance between deletion and duplications breakpoints? Does the model distinguish these two types?
5) When splitting the training/validation/testing dataset, there will be significant data leakage if this is done randomly. Many SV/CNV regions have characteristic repeats patterns.
6) The comparison between CNV-Net and other methods is not fair as the other methods are finding CNVs in the whole genome while CNV-Net is given a pre-defined set of "candidate breakpoint regions" in which positive samples have breakpoints perfectly centered in the middle. Moreover, such candidate regions might suffer from data leakage.
7) Does HG002 really only has 174 duplications while NA12878 has 22,936?
8) GIAB has more than two genomes available. Can we test on more genomes?
9) Minor: RELU -> ReLU

---

### Official Review · AnonReviewer2 · 2020-10-29
**Interesting approach, but more evaluation is needed**

**Rating:** 5
**Confidence:** 3

**Review:**

The authors present an innovative approach to CNV detection using CNNs. However, I believe that additional experiments need to performed before this paper is ready for publication. Below is a breakdown of strengths and weaknesses of this submission.

Strengths:
1. The empirical results seem encouraging compared to baselines.
2. The method seems easy to implement and runs very fast.

Cons:

1. There is only one comparison performed. I was wondering what the results are like in the opposite direction (training in HG002 and testing on NA12878)? Furthermore, the authors should consider other datasets, in particular the TCGA data where bam files, and copy number variations are available. The authors can train the method on one patient and test on another.

2. There is lack of analysis on whether the method can recover known canonical copy number variations. In the TCGA dataset, and for certain cancer types, there are small and large copy-number variations that are known and have been validated in the lab. The authors should ensure that their method can recover these events using their method.

3. The authors have considered baselines that do not take as input already known break points, which makes the comparison somewhat unfair. I was wondering if there is a way in which the authors can use the discovered breakpoints for each respective baseline and then determine whether their algorithm can improve the predictions?

---

### Decision · Program_Chairs · 2021-01-07
**Final Decision**

**Decision:**

Reject

**Comment:**

Four knowledgeable referees have indicated reject. I agree with the most critical reviewer R4 that the model design lacks a clear and transparent motivation and that the experimental setup is not convincing, and so must reject.